# The Relationship between Sperm Oxidative Stress Alterations and IVF/ICSI Outcomes: A Systematic Review from Nonhuman Mammals

**DOI:** 10.3390/biology9070178

**Published:** 2020-07-21

**Authors:** Jordi Ribas-Maynou, Marc Yeste, Albert Salas-Huetos

**Affiliations:** 1Biotechnology of Animal and Human Reproduction (TechnoSperm), Institute of Food and Agricultural Technology, University of Girona, 17003 Girona, Spain; marc.yeste@udg.edu; 2Unit of Cell Biology, Department of Biology, Faculty of Sciences, University of Girona, 17003 Girona, Spain; 3Andrology and IVF Laboratory, Division of Urology, Department of Surgery, University of Utah School of Medicine, Salt Lake City, UT 84108, USA

**Keywords:** infertility, sperm, oxidative stress, ROS, DNA damage, IVF, ICSI

## Abstract

Achieving high embryo quality following IVF and ICSI procedures is a key factor in increasing fertility outcomes in human infertile couples. While the male factor is known to underlie infertility in about 50% of cases, studies performed in human infertile couples have not been able to define the precise effect of sperm affectations upon embryo development. This lack of consistency is, in most cases, due to the heterogeneity of the results caused by the multiple male and female factors that mask the concrete effect of a given sperm parameter. These biases can be reduced with the use of animal gametes, being a good approach for basic researchers to design more homogeneous studies analyzing the specific consequences of a certain affectation. Herein, we conducted a systematic review (March 2020) that assessed the relationship between sperm oxidative stress alterations and IVF/ICSI outcomes in nonhumans mammals. The review was conducted according to PRISMA guidelines and using the MEDLINE-PubMed and EMBASE databases. Thirty articles were included: 11 performed IVF, 17 conducted ICSI, and two carried out both fertilization methods. Most articles were conducted in mouse (43%), cattle (30%) and pig models (10%). After IVF treatments, 80% of studies observed a negative effect of sperm oxidative stress on fertilization rates, and 100% of studies observed a negative effect on blastocyst rates. After ICSI treatments, a positive relationship of sperm oxidative stress with fertilization rates (75% of studies) and with blastocyst rates (83% of studies) was found. In conclusion, the present systematic review shows that sperm oxidative stress is associated with a significant reduction in fertilization rates and in vitro embryo development.

## 1. Introduction

Infertility is defined as the failure to achieve a clinical pregnancy after 12 months of regular unprotected intercourse [1]. In order to help human infertile couples, assisted reproductive techniques (ART) were developed decades ago; nowadays, all intrauterine insemination (IUI), in vitro fertilization (IVF) and intracytoplasmic sperm injection (ICSI) are methods with the potential to result in a pregnancy and a healthy newborn. Because of that, they are currently accepted worldwide as having a routine medical use. According to the latest report from the European Society of Human Reproduction and Embryology (ESHRE) [2], about 729,000 ART treatments are performed per year in Europe. Among them, 25.8% correspond to IUI, 21.4% to IVF and 52.8% to ICSI. Thus, procedures involving artificial fertilization (i.e., IVF or ICSI) are the most frequent ART, with more than 541,000 cycles a year. Despite this, large numbers of couples resort to IVF and ICSI every year, and clinical pregnancy rates are still far from being considered fully effective, ranging between 26.0% and 28.5% per cycle [2]. These results show that there is still room for the improvement with these procedures. Therefore, it is necessary to gain new insights into the factors that lead to the production of embryos of high quality and with good implantation chances [3,4].

ART cycle outcomes and embryo quality are defined by different factors, such as embryo genetics, kinetics and morphology, and fertilization and blastocyst rates, amongst others. Regarding embryo genetics, the discovery and application of preimplantation genetic screening (PGS) effectively helped identify embryos with euploid genetic complement [5]. Implementing this procedure has led to a global increase in pregnancy rates, and nowadays, it is applied to more than 40% of cycles in some countries through next generation sequencing [6]. As far as embryo kinetics are concerned, not only does time lapse imaging help in identifying embryos with normal development kinetics, but analyzing such kinetics has been purported to be a useful, noninvasive test that may increase pregnancy rates [7]. Despite our current ability to select euploid embryos with normal morphokinetics, in some cases, we are not able to determine the etiologies affecting fertilization rates that lead to a lower embryo number and compromise pregnancy and blastocyst rates [8,9]. In this context, study of the male factor, which contributes to half of the couple’s disease, is important if we are to understand the disorder affecting a specific couple. In this regard, and despite a huge effort having been made to understand the reasons for male infertility and the reduction of sperm quality, the clinical effects of some male parameters to ART outcomes remain unsolved issues or controversial topics [10]. It is well known that one of the major causes of a reduction of sperm quality is oxidative stress, which is related to an imbalance of reactive oxygen species (ROS). ROS are oxygen-derivate molecules with high reactivity due to their free electrons or free radicals, potentially causing a wide variety of modifications to different molecules. Low levels of ROS are physiologically necessary for sperm capacitation and sperm–oocyte interaction [11,12], and therefore, for sperm fertilizing ability. However, increases of ROS are highly damaging when they are not mitigated by the antioxidant mechanisms of the sperm cell and/or seminal plasma. The damaging effects caused by ROS are lipid peroxidation, which alters plasma membrane permeability and fluidity [13], protein modifications, reduction of ATP production [14] and sperm DNA damage, which entails base modifications such as 8-OH-guanine or 8-OH-2′ deoxyguanosine. As spermatozoa are lacking DNA transcription and translation, and the mentioned effects cannot be counteracted, DNA alterations cannot be repaired [15]. Therefore, these effects lead to a general reduction of sperm quality, expressed as loss of motility and decreased DNA integrity.

Over the last decade, different studies have aimed at determining the clinical effects of low sperm quality upon ART outcomes, both in humans [16,17,18] and animals [19,20]. This research has demonstrated that low sperm motility and high sperm DNA fragmentation have a detrimental impact on natural fertility, reducing pregnancy achievement and increasing pregnancy loss [16,21,22]. Despite this clear consensus about the negative effects of sperm quality on natural fertility, studies analyzing the male factor influence on IVF and ICSI outcomes in humans have reported conflicting results. In this regard, some works have proven that low sperm quality leads to a reduction of pregnancy rates, whereas others did not find such an association [23,24,25,26]. Whilst the most recent meta-analyses published on the topic did not resolve this question, they pointed out that sperm quality may have more influence on IVF than on ICSI outcomes, probably due to the sperm selection performed in the latter [27,28]. In addition, most previous studies concur on the high heterogeneity of studying human infertility disorders, regarding both intrinsic factors and the heterogeneity found within couple factors [29,30]. Thus, the heterogeneities caused by these factors are the main problem in study designs for human subjects, since they mask the single male factor that researchers specifically aim to study. Because of that, focusing research on animal gametes makes it possible to design more homogeneous experiments, reducing female heterogeneity by using a single or a few animal donors, and reducing male heterogeneity by inducing the concrete damage that the researcher seeks to investigate. In animals, assisted reproduction technologies, such as artificial insemination, is widespread in pigs, cattle, sheep and horses [31,32,33]. In contrast, IVF and ICSI are not usually performed, but are utilized for research or other specific purposes, such as embryo production, sex selection and genetic selection [34,35,36]. In research, the induction of DNA damage to healthy biological samples and the homogeneity of sperm and oocyte samples make it possible to design more homogeneous studies than those performed in infertile humans [19,37,38]. Therefore, experiments mimicking the sperm oxidative damage happening naturally in human infertile males are important to reach firm conclusions regarding the effect of this damage to ART. The aim of the present study is to systematically review studies performing induction of sperm oxidative damage in mammals other than humans in order to elucidate the influence of the male factor on IVF/ICSI outcomes.

## 2. Methods

### 2.1. Systematic Review Execution and Registration

Preferred Reporting Items for Systematic Reviews and Meta-Analyses (PRISMA) guidelines [39] were followed in this systematic review. The protocol used was registered in the international prospective register of systematic reviews (PROSPERO 2020: CRD42020176683).

### 2.2. Data Sources and Systematic Search Strategy

The inclusion and exclusion criteria defined in the PICOS (Population, Intervention, Comparator, Outcome, Study) design structure (Appendix A) led to a comprehensive list of keywords and Medical Subject Headings (MeSH) terms, which were combined with keywords related to oxidative stress and IVF/ICSI outcomes, all them resulting in the design of the search strategy (Appendix B). The systematic literature search, performed with both MEDLINE-PubMed (www.ncbi.nlm.nih.gov/pubmed) and EMBASE (www.embase.com/#search) databases, included articles from the earliest date and until 27 March 2020.

### 2.3. Study Eligibility

The eligibility criteria for the studies included in the present systematic review was defined prior to the search in the PICOS design structure (Appendix A). Research that fulfilled the following criteria was selected to be included in the review: (i) studies performed on mammals other than humans; (ii) studies aimed at elucidating the effect of sperm oxidative affectation on IVF and ICSI; (iii) studies in which the induced sperm oxidative affectation was measured and compared to controls; and (iv) primary outcomes were defined as IVF/ICSI cycle parameters, such as fertilization and blastocyst rates. Alternatively, studies that performed natural mating or IUI, those that did not measure the effect of oxidative treatments on sperm cells, and studies aimed at analyzing the protective effect of components added to cryopreservation/thawing media were excluded from this review. Review articles, systematic reviews/meta-analysis, case reports, letters and commentary articles were also considered as noneligible.

### 2.4. Study Selection Procedure

As shown in Figure 1, the study selection procedure was conducted in different stages and following the aforementioned eligibility criteria. First, the search was performed in both the PubMed and EMBASE databases using a standardized extraction form that collected the following information: reference, digital object identifier (DOI), publication year, title, abstract, authors, article type, affiliations, and the database in which it was found. Once all records were annotated in the database, duplicate articles and those declared as noneligible were excluded. The second step in the study selection was based on a screening according to the title and abstract, excluding articles that did not meet the eligibility criteria. This step was performed by two researchers that are specialists in male (in)fertility (J.R.-M. and A.S.-H); in case of disagreement, a third author (M.Y.) was involved. Afterwards, full texts of all selected articles were downloaded and screened for a third phase of exclusion conducted in parallel by the same two authors. Discrepancies regarding eligibility were discussed with the third author (M.Y.) to reach a consensus. The exclusion reasons are depicted in Figure 1.

### 2.5. Data Extraction for Systematic Review and Statistics

After completing the eligibility process, the final list of articles included in the systematic review was analyzed in depth to extract the following data: reference, aim of the study, animal species used, treatment applied to induce oxidative damage in sperm cells, sperm parameters measured, effects caused by the treatment to sperm cells, origin of the oocytes used in IVF or ICSI, fertilization procedure (IVF, ICSI or both), effects observed on IVF/ICSI outcomes and conclusion of the study.

During the data extraction process, the studies included in the systematic review were classified according to whether or not their findings supported the fact that sperm oxidative affectation led to a reduction of fertilization rates, blastocyst rates, and implantation and pregnancy rates. These data were divided regarding the fertilization method used (IVF or ICSI), and the results were compared using the Chi-squared test (Statistical Package for the Social Sciences Ver. 25.0, SPSS, IBM, New York, NY, USA).

## 3. Results

### 3.1. Identification and Selection of Articles for Qualitative Analysis

Figure 1 shows the flowchart used to conduct the search and study selection process. The initial search gave rise to the identification of 1451 articles in the MEDLINE-PubMed database, and 2191 in the EMBASE database, resulting in 3642 records. After including all the records in a database, 47 articles were removed because they were duplicated and 220 were removed because they were reviews, and therefore, did not meet the inclusion criteria. This first database clean up led to the further screening of 3375 articles by title and abstract, performed by two researchers (J.R.M. and A.S.H.). This resulted in the exclusion of 3295 studies, which were performed on humans, presented irrelevant outcomes or did not meet the scope of the present systematic review; the 80 remaining articles were downloaded and analyzed in parallel to determine their eligibility for inclusion according the same criteria. Fifty studies were excluded for the following reasons: 16 did not use IVF or ICSI fertilization methods, 13 did not analyze oxidative damage caused to sperm cells, 12 had outcomes that were irrelevant to the scope of the review, seven studied the effect of additives or extenders to the cryopreservation process, one compared two treated groups but lacked a control group, and one was performed on human subjects. Therefore, after applying all the eligibility parameters, 30 articles remained for qualitative analysis.

### 3.2. Systematic Review: Qualitative Analysis

The 30 studies included in the systematic review were carefully analyzed in order to extract the key data summarizing: (a) their aims; (b) the treatments performed to induce sperm oxidative damage; and (c) the IVF/ICSI outcomes obtained as a result of the treatment, as described in the methods section. This analysis is summarized in Table 1 and Table 2. Among the studies, 11 (36.6%) used IVF as a fertilization method, 17 (56.6%) used ICSI, and two (6.6%) used both techniques. Regarding the animal models used, three studies (10%) were performed on pigs, nine (30%) were conducted on cattle, 13 (43.3%) were performed on mice, one (3.3%) compared mice and hamsters, one study (3.3%) was conducted on sheep, one (3.3%) was carried out on rats, one (3.3%) was conducted on horses and one (3.3%) was performed on macaques. The percentage of studies performing IVF or ICSI differed among different animal models. All studies in pigs (3/3) performed IVF; for cattle, 89% (8/9) of studies conducted IVF and one study performed ICSI; and for mice, 54% (7/13) of studies conducted IVF and 46% (6/13) carried out ICSI.

Regarding the origin of the oocytes used, 37% (11/30) of the studies collected the ovaries and performed in vitro maturation (IVM), 53% (16/30) conducted superovulation, one recovered oocytes from induced ovulation and two did not provide this information. These origins also differed between species; thus, whereas porcine and bovine studies mainly used IVM oocytes (100% and 78% of the studies, respectively), superovulation was the main oocyte source in studies on mice (92% of the articles).

#### 3.2.1. Effect of Different Treatments on Spermatozoa

A very wide variety of treatments were tested in different studies. Eighteen studies applied treatments to sperm cells, nine applied treatments to animals or to their testis, and three performed comparisons between cohorts of animals with high and low fertility. With regard to studies performing treatments, three applied heat stress, three applied radiation (X rays or Gamma rays), seven treated sperm with oxidant molecules, such as H_2_O_2_ or xanthine oxidase, one applied genetic modifications creating mouse chimeras for Protamine 2 (*Prm2*) gene, one caused damage through freeze-thawing, and eleven performed treatments with other chemical agents, such as manganese and calcium, lipopolysaccharide, cyclophosphamide, alkaline pH, fluoride, pyridaben, and DAVA, or fed animals with a selenium-deficient diet.

Most of these treatments had a detrimental effect upon sperm quality, and only three studies observed no effect on the evaluated parameters. Hence, DNA damage occurred in 17 out of 19 studies analyzing this parameter; ROS production and antioxidant capacity were affected in all of the studies that took these two indicators into consideration (eight and two studies, respectively), and five out of six studies observed affectations related to plasma membrane oxidation, such as lipid peroxidation.

#### 3.2.2. Effect of the Different Treatments on IVF/ICSI Outcomes

All the included studies aimed to analyze the effects of sperm alterations on IVF/ICSI outcomes, such as fertilization, blastocyst, implantation and live birth rates. Taking IVF and ICSI treatments together, 78% of the studies analyzing fertilization rates (21/27) found an association between sperm oxidation and this parameter, and 92% of the studies analyzing blastocyst rates (22/24) found that it was related to sperm oxidation. Considering only the works that performed IVF, 80% (12/15) found a relationship between sperm oxidation and fertilization rates, while 100% (12/12) found a negative association with blastocyst rates. Studies performing ICSI displayed a similar trend: 75% (9/12) found a relation to fertilization rates and 83% (9/12) found an association with blastocyst rates. Oxidative stress affected IVF and ICSI treatments in a similar percentage, and no differences between fertilization methods were found, either for fertilization rate (*p* = 0.756) or for blastocyst rate (*p* = 0.140).

Among all the studies included in this systematic review, only two [53,60] using mice analyzed implantation rates after ICSI procedures, finding a reduction of this parameter. Li et al. [53] and Li and Lloyd [54] also reported similar findings regarding live birth rates; these studies were the only ones which analyzed this parameter.

## 4. Discussion

Thanks to several studies analyzing male factor infertility performed by multiple research groups around the globe, the male partner has gathered more attention in routine clinical practice. Nowadays, it is well known that the male factor contributes equally to the female factor to infertility disorders that affect nearly 50 million couples around the world [67]. The present study systematically reviewed the effects of male oxidative damage on the IVF and ICSI outcomes analyzed in animal models, showing that oxidative affectations in sperm are translated into poorer outcomes after ART. The most affected parameters were fertilization rates and blastocyst rates, since male oxidative damage induces an arrest during embryo cleavage.

Different studies have proven that oxidative stress represents an important issue leading to sperm cell defects such as DNA damage, protein modifications, plasma membrane damage and even epigenetic modifications that can be transmitted to the offspring [68,69]. Oxidative DNA damage in mature sperm cells is irreparable, since the nature of the DNA condensation impedes the transcription of DNA repair enzymes; therefore, the integrity of the genetic material is compromised in the male gamete [15]. Also, cell membranes are particular susceptible to oxidative stress due to the presence of polyunsaturated fatty acids. Free radicals react with fatty acids, producing highly reactive aldehydes which are capable of inhibiting antioxidants, such as glucose-6-phosphate dehydrogenase (G6PDH) and glutathione peroxidase (GPX), and also leading to motility loss [13,70]. All these effects represent a reduction in sperm quality, which has been related to increased difficulty achieving natural pregnancy and to male infertility in a number of studies [16,17,21]. Classical sperm analyses, including concentration, motility and morphology, are among the key initial tests in defining male infertility, while sperm DNA damage represents a more advanced test that may help in defining the physiopathology of the male partner [18,23].

Alternatively, oxidative stress also has effects at different levels. Regarding the sperm epigenome, it has been demonstrated that oxidative stress induces tRNA cleavage [71], generating tsRNA and rRNA fragments that are involved in the sperm-mediated inheritance of intergenerational pathologies [68,72]. Moreover, recent insights also indicated that cigarette smoking can induce sperm oxidative stress, generating DNA methylation changes [73]; this suggests that oxidative stress is an important mechanism through which an unhealthy lifestyle may impact the offspring phenotype [74]. In fact, some authors described that the expression of some sperm small RNAs and tsRNAs rapidly changed in relation to male high-sugar diets [75,76]. The profiles of these RNA were related to sperm quality and embryo viability [77], and may also affect histone modifications and tsRNA biogenesis at the testis level [78]. Therefore, epigenetic components are important affectations to be studied when sperm cells are subjected to an oxidative environment.

Assisted reproductive methods such as IVF and ICSI are commonly applied to infertile couples, representing a good alternative to overcome some infertility issues and achieve pregnancy, despite their limited global effectiveness [2]. IVF and ICSI rely on the generation of good quality embryos to be selected and transferred into the female uterus in order to achieve a clinical pregnancy. While oocyte quality is widely known to be crucial for embryo development [79], the data are inconsistent with regard to the relationship between sperm quality and IVF and ICSI outcomes in humans. In search for a consensus, the most widespread conclusion in the most recent meta-analyses is that while DNA damage may have an influence on pregnancy achievement after IVF treatments, this influence has not been clearly established after ICSI procedures [27,28]. These differential results may indicate that the single sperm selection performed in ICSI may be related to ART outcomes, but resulted in a controversy between different reproductive associations regarding the use of sperm DNA fragmentation as a clinical routine test [80,81]. The meta-analyses performed in humans agree with the high heterogeneity of the results, mainly given by the variety of female factors affecting the cycle, the technicians performing the ART techniques, the methodology used for sperm DNA damage analyses, and the different grades of male factor affectations, amongst others [25,27,28]. Testing the effect of the male factor through animal models may help reduce this heterogeneity, since they make it possible to induce precise and homogeneous oxidative damage to sperm cells, and a cohort of homogeneous oocytes can be also obtained. In this sense, despite the application of different treatments among studies, an effect on sperm cells was found in almost 90% of the studies. In this systematic review, we found that these effects at the sperm level are translated into affectations in fertilization rates and blastocyst rates, which is supported by more than 80% of studies on IVF and more than 75% of the studies on ICSI. In addition, these effects were proven for different animal species (including rodents, cattle, pig, sheep and macaque) and for different oocyte origins (IVM or superovulated), showing that the sperm quality in terms of oxidative damage was highly related to the fertilization and blastocyst rates. Interestingly, the findings reported herein are supported by Toyoshima, 2009 [82], who indicated that a P53 and P21-related DNA damage response (S-phase and G2/M) occurs during early embryo development. These responses are activated at different time points; while a p53 dependent S-phase checkpoint is active from the first mitosis to ensure normal embryo division, the G2/M checkpoint is activated before blastocyst, causing an arrest of affected embryos [82,83]. Thus, as DNA breaks caused by irradiation compromise embryo development, oxidative damage may also have an important role in embryo progression.

### 4.1. Strengths and Limitations

The present systematic review provides a new overview of the relationship between sperm oxidative damage and IVF/ICSI results. To the best of our knowledge, all systematic reviews and meta-analyses performed until the present moment have been focused on human patients; they observed a high heterogeneity in their results, probably due to individual differences attributable to both male and female factors. Here, we place the focus on reviewing studies in animal models (laboratory or production animals), which makes it possible to design more homogeneous studies due to the use of a single sperm sample with different treatments to inseminate multiple oocytes, usually obtained from the same female or group of females. This reduces the presence of confounding factors, and elucidates the effect of the treatment.

In spite of this, as a main limitation, our review included studies performed on different animals, where different oocyte origins had been used. While most studies performed on cattle, pigs and sheep used IVM oocytes, most studies performed on rodents used superovulated oocytes, which represents a source of bias in comparison to human studies. The searches performed using both MEDLINE-PubMed and EMBASE databases were designed to identify all potentially eligible studies following the criteria previously defined in the PICOS table (Appendix A). However, some small biases may potentially exist if a study aimed at inducing damage other than oxidative stress performed a treatment on animals or sperm that led to oxidative damage as an unintended consequence.

Additionally, our study is not exempt from publication biases. Studies that did not find an association between the studied parameters would be less likely to get published, and therefore, may be underrepresented in scientific databases. Also, the eligibility of articles published during the period between the search and the publication of the present paper was not assessed.

### 4.2. Conclusions

Notwithstanding the controversies existing in human studies regarding the impact of sperm alterations on IVF and ICSI outcomes, the present systematic review supports the hypothesis that, in animals, oxidative affectations at the sperm level decrease fertilization and blastocyst rates. Then, a cause-effect of sperm cells is observed in embryos. Finally, despite the fact that most studies did not assess implantation and pregnancy rates, the data included in our review suggest that these parameters might also be compromised in a scenario where oxidative sperm damage is present.

## Figures and Tables

**Figure 1 biology-09-00178-f001:**
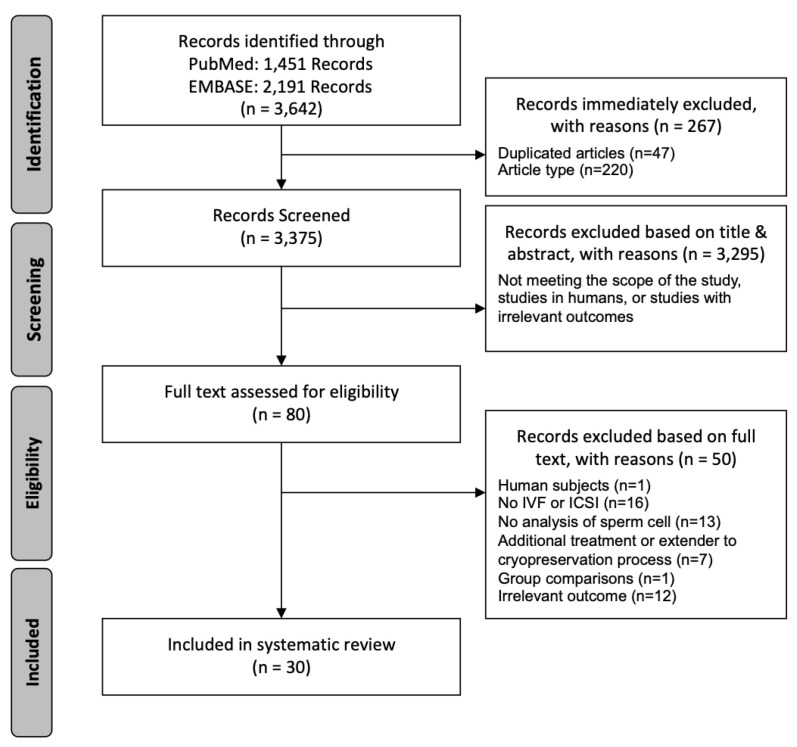
Flowchart of the literature search and selection process.

**Table 1 biology-09-00178-t001:** Summary of the fertilization methods and oocyte origin regarding different animals.

Animal	Total Studies	Fertilization Method		Oocyte Origin		
		IVF Studies	ICSI Studies	IVM	Superovulated	Nonspecified
Pig	3	3	0	3	-	-
Cattle	9	8	1	7	1	1
Mice	13	7	6	-	12	1
Mouse and Hamster	1	0	1	-	1	-
Sheep	1	0	1	1	-	-
Rat	1	1	0	-	1	-
Macaque	1	0	1	-	1	-
Horse	1	0	1	-	1 (only ovulation induction)	-

**Table 2 biology-09-00178-t002:** Summary of the studies included in the systematic review.

ID	Reference	Aim	Animal	Treatment	Sperm Parameter Measured	Effects of Treatment on Sperm	Oocyte Origin	IVF or ICSI	Effects on IVF/ICSI	Conclusion
1	Ahmadi et al. 1999 [40]	To study the developmental capacity of spermatozoa at various levels of disintegration.	MouseHamster	Frozen twice without cryoprotectant plus DTT and gamma radiation	DNA damage: TUNEL	Increased DNA damage.	Super-ovulated	ICSI	- Similar fertilization rates - Lower blastocyst development - Lower live birth rate	Impairment of sperm cells causes a reduction in ICSI outcomes.
2	Arias et al. 2014 [41]	To evaluate the effects of pretreatment of bovine spermatozoa with NaOH and DTT on in vitro developmental potential.	Cattle	Incubation of sperm with 5 mM DTT for 20 min and 1 mM NaOH for 60 min	DNA fragmentation: TUNEL	Highest concentrations of NaOH led to high DNA damage.	Super-ovulated	ICSI	- High concentration of NaOH blocked blastocyst development	NaOH has a detrimental effect on embryo development after ICSI.
3	Bittner et al. 2018 [42]	To elucidate whether sperm oxidative stress results in increased DNA damage in the embryo.	Cattle	Incubation of sperm with H_2_O_2_	DNA damage: SCSA	Exposure of sperm to H_2_O_2_ led to an increase of DNA damage in the embryo.	IVM	IVF	- Embryo development was delayed when sperm were incubated with H_2_O_2_ treatment - Reduced cleavage and blastocyst rates	Oxidative stress in spermatozoa induces developmental abnormalities in the embryo.
4	Burruel et al. 2013 [38]	To determine whether sperm oxidative damage induced by ROS affects embryo development.	Rhesus Macaque	Oxidation of sperm cells with xanthine oxidase.	- Lipid peroxidation	Treatment with xanthine oxidase led to an increase of lipid peroxidation.	Super-ovulated	ICSI	- Exposure to high levels of ROS leads to arrest of embryo development before eight-cell stage. Additionally, all stages were affected in the treated group.	Paternal oxidative stress influences early embryo development.
5	Castro et al. 2016 [43]	To assess the impact of sperm oxidative stress on embryo development.	Cattle	Incubation of sperm with increasing doses of H_2_O_2_	- Oxidative status: CellROX green - DNA damage: SCSA	Incubation causes an increase of oxidative stress and alterations in chromatin integrity.	IVM	IVF	- Cleavage rates, development to 8 cells and percentages of blastocysts are inversely related to H_2_O_2_ concentration	Oxidative environment can impair bull sperm quality, affecting embryo development.
6	Castro et al. 2018 [44]	To predict in vitro bull fertility with the assessment of DNA integrity.	Cattle	High vs. low fertility bulls based on embryo development rate	- DNA damage: SCSA and Comet - DNA condensation: CMA3	None.	Not specified	IVF	- SCSA and Comet did not show differences between high and low fertility groups - DNA condensation (CMA3 test) was lower in the high fertility group.	The differences found in DNA condensation through CMA3 were not sufficient to explain differences in fertility rates.
7	Cho et al. 2003 [45]	To determine whether a reduction in protamine content results in failure to transmit the male genome to next generation.	Mouse	Chimeric, *Prm2*^−/>−^ mice	- DNA damage: Comet assay	DNA damage was observed in chimeras, affecting more than 70% spermatozoa.	Not specified	ICSI	- Arrested embryos at metaphase. - Less embryos developed to later stages.	Protamine 2 is essential to maintain sperm DNA integrity and to promote embryo development.
8	Ebadi Manas et al. 2013 [46]	To elucidate how pyridaben can affect sperm quality and in vitro fertilizing ability.	Mouse	Oral administration of pyridaben for 45 days at two different doses	- Chromatin integrity: Aniline blue and acridine orange	Increase of sperm DNA damage.	Super-ovulated	IVF	- Lower fertilization rates - Lower development to blastocyst rates	Pyridaben induces DNA damage, which decreases fertilization and blastocyst rates.
9	Fatehi et al. 2006 [47]	To investigate whether and at what level the paternal DNA damage influences fertilization and embryo development.	Cattle	Irradiation with X or Gamma rays	- DNA damage: TUNEL and Acridin Orange	Sperm DNA damage increased in a dose dependent manner.	IVM	IVF	- Similar rates of embryo cleavage at day 4 - Embryo development to blastocyst was severely impaired	DNA damage induced by radiations impair embryo development to blastocyst.
10	Gawecka et al. 2013 [19]	To test how zygotes respond to DNA damage during the first cell cycle.	Mouse	Induction of DNA damage with manganese and calcium	- DNA damage: Pulsed Field Gel Electrophoresis	Double strand DNA damage detected by Pulsed Field Gel Electrophoresis (PFGE).	Super-ovulated	ICSI	- Chromosomal alterations at first paternal pronucleus - No fertilization impairment - Embryo development was delayed after fertilization and arrested before reaching the blastocyst stage	DNA damage is important for proper embryo development at initial stages.
11	Gonçalves et al. 2010 [48]	To investigate the effects of antioxidants and a pro-oxidant on the quality and fertilizing ability of bull spermatozoa.	Cattle	Sperm were co-incubated with antioxidants (β-mercaptoethanol and Cysteamine) or pro-oxidant (buthionine sulfoximine) molecules	- DNA damage: Acridine Orange	No effect on sperm DNA damage, but reduction of plasma membrane integrity.	IVM	IVF	- Antioxidants and pro-oxidants reduced fertilization and blastocyst rates	Supplementation with antioxidants and with pro-oxidants during IVF procedures impaired sperm quality, normal pronuclear formation and development to blastocyst.
12	Gonzalez-Castro et al. 2018 [49]	To determine which sperm population characteristics are predictive of ICSI outcome.	Horse	Evaluation of DNA damage and plasma membrane integrity	- Membrane integrity: HOS test - DNA damage: SCD test	None.	Ovulation induction and follicle aspiration	ICSI	- DNA damage was not associated to any fertilization, embryo development or pregnancy	Plasma membrane integrity was the parameter with higher association to ICSI outcome
13	Hourcade et al. 2010 [50]	To investigate if female tract is able to select nonaffected sperm cells.	Mouse	Heat stress and Gamma radiation to animals.	- DNA damage: Comet assay	The extent of DNA damage in sperm obtained from female tract was higher following heat stress than after gamma radiation.	Super-ovulated	IVF and ICSI	- Heat stress reduced fertilization rates in both IVF and ICSI - Gamma radiation decreased blastocyst rates in gamma radiation in both IVF and ICSI	- Sperm DNA fragmentation affects IVF and ICSI. - ICSI should be performed with highly motile sperm.
14	Izquierdo-Vega et al. 2008 [51]	To evaluate the effects of fluoride on in vitro sperm fertilizing ability.	Rat	Rats administered with 5 mg fluoride/kg/body mass/24 h for eight weeks	- SOD activity - Intracellular superoxide anion levels - Lipid peroxidation	Fluoride increases oxidative stress, decreases the antioxidant activity of SOD, and increases lipid peroxidation.	Super-ovulated	IVF	- Lower ability to fertilize oocytes	Fluoride exposure causes a decrease in sperm fertilization capacity.
15	Jang et al. 2010 [52]	To examine the developmental rates of semen treated with and without melatonin in the presence of H_2_O_2_.	Pig	Treatment of semen with H_2_O_2_ with and without melatonin.	- Lipid peroxidation: Malondialdehyde levels	Exposure to H_2_O_2_ augmented lipid peroxidation, but this increase was mitigated by melatonin.	IVM	IVF	- Fertilization was not impaired by exposure of sperm to H_2_O_2_ - Exposing sperm to H_2_O_2_ arrested the development to blastocyst, but melatonin prevented that negative effect.	Supplementation with melatonin could improve sperm quality, increasing the developmental capacity of porcine embryos.
16	Li et al. 2009 [53]	To evaluate whether normal offspring can be generated after exposing sperm to high NaOH concentrations.	Mouse	High alkaline treatment of sperm cells	- DNA damage: Acridine Orange	Increase of DNA damage.	Super-ovulated	ICSI	- Decrease of fertilization rates - Reduction of implantation and live pups born - Increase of chromosomal damage in embryos	Sperm treated with NaOH retain their ability to activate the oocyte, but embryo development is lower as NaOH concentration increases.
17	Li et al. 2020 [54]	To assess the extent to which the measurement of DNA fragmentation index in sperm can predict their fertilizing ability.	Mouse	Freeze-thawing	- DNA damage: TUNEL	Cryopreservation increased DNA damage.	Super-ovulated	IVF and ICSI	- Sperm DNA damage reduced fertilization rates following both IVF and ICSI. - Samples with higher DNA damage resulted in a reduction of viable offspring.	DNA fragmentation index is an accurate parameter to determine sperm quality and fertility potential.
18	Llamas Luceño et al. 2020 [55]	To address the impact of natural heat stress on bull fertility.	Cattle	Comparison of different bulls exposed to separate levels of heat stress	- ROS production - Lipid peroxidation - DNA fragmentation: TUNEL	No differences in H_2_O_2_ production, lipid peroxidation and DNA damage. A tendency to significance in total ROS production was observed.	IVM	IVF	- No impact of exposing sperm to heat stress upon cleavage rates - A reduction of day 7 and 8 blastocyst rate and a delay of blastocyst hatching were observed.	Sperm fertilizing ability decreases due to male exposure to heat stress.
19	Makvan-di et al. 2019 [56]	To show the potential benefits of alpha lipoic acid as antioxidant in lipopolysaccharide- treated sperm cells.	Mouse	Sperm cells were incubated with lipopolysaccharide and alpha lipoic acid at different concentrations	- ROS production: DCFH-DA test - DNA damage: Acridine orange	Lipopolysaccharide augmented ROS and DNA damage, but the addition of alpha lipoic acid mitigated those increases.	Super-ovulated	IVF	While lipopolysaccharide led to lower fertilization, cleavage, compaction and blastocyst rates, this effect was reverted by alpha lipoic acid.	Alpha lipoic acid is a strong antioxidant and protective sperm factor.
20	Matini Behzad et al. 2014 [57]	To evaluate the effects of fish oil feeding on sperm parameters and its incidence in IVF.	Sheep	Fish oil diet for 70 days	Intracellular ROS	Superoxide anion was lower in the treated than in the control group.	IVM	IVF	- Cleavage rates were higher in the treated group.	Addition of fish oil to ram diet improves sperm quality and in vitro fertilization ability.
21	Mehraban et al. 2019 [58]	To evaluate the antioxidant effects of Gallic Acid on apoptotic-like changes in sperm.	Mouse	Cyclophosphamide and gallic acid	Apoptotic-like changes: Annexin V staining	Cyclophosphamide induces apoptotic-like changes compared to controls. Gallic Acid mitigates that increase.	Super-ovulated	IVF	- Cyclophosphamide reduces fertilization rates and proportions of cleaved embryos.	Gallic Acid suppresses ROS induced by Cyclophosphamide and help rescue fertility.
22	Paul et al. 2008 [59]	To explore the link between heat stress, DNA damage and the impact on cell function.	Mouse	Acute testicular heat stress at different temperatures	- Sperm DNA damage: SCSA - Spermatocyte DNA damage	Increase of DNA damage in spermatocytes and spermatozoa when heat stress was 40 °C/42 °C.	Super-ovulated	IVF	- Fertilization was similar between groups. - Development to 2-4 cells was similar to controls. - Development to blastocyst was severely reduced.	Spermatogenesis is impaired when scrotal temperatures increase, showing that exposing male to high temperatures reduces embryo development and pregnancy of the oocytes fertilized with those sperm.
23	Perez-Crespo et al. 2008 [60]	To evaluate whether factors released from membrane-damaged spermatozoa have a role in DNA damage and ICSI outcomes.	Mouse	Incubation of sperm in media containing factors released by damaged sperm	- DNA damage: TUNEL	Increase of sperm DNA damage.	Super-ovulated	ICSI	- Cleavage and blastocyst rates were not impaired by treatment - Implantation was reduced, regardless of whether embryos were transferred at 2-cell stage or at blastocyst stage.	Factors released from membrane-damaged sperm are capable of inducing DNA fragmentation in viable spermatozoa, and decrease implantation rates.
24	Sanchez-Gutierrez et al. 2008 [61]	To evaluate the effect of selenium deficiency on in vitro fertilizing ability.	Mouse	Selenium deficient diet for 4 months	- Glutathione peroxidase activity - Lipid peroxidation	Glutathione peroxidase activity was reduced and lipid peroxidation was increased in selenium free diet.	Super-ovulated	IVF	- Fertilization rates were reduced	Selenium deficiency leads to a reduction of sperm quality and fertilizing ability.
25	Silva et al. 2007 [62]	To describe the effects of exposing bull sperm to mild and intense ROS generation conditions.	Cattle	Sperm were incubated with pro-oxidant molecules	- Sperm DNA oxidation - Lipid peroxidation - Cytosol and mitochondrial oxidation	Increased intracellular and mitochondrial ROS, while no significant increase of DNA oxidation was observed.	IVM	IVF	- Reduced cleavage rates and blastocyst rates when high oxidation was applied.	Oxidative stress can result in damaged sperm cell structures, affecting embryo development.
26	Simoes et al. 2013 [63]	To evaluate the influence of sperm oxidative stress susceptibility on DNA fragmentation and IVF outcomes.	Cattle	Division in four groups according to oxidative stress susceptibility	- DNA damage: Comet assay, Acridine Orange and TUNEL	Sperm DNA was compromised in response to increased oxidative stress susceptibility.	IVM	IVF	- Embryo cleavage decreased as oxidative stress increased - No significant differences in blastocyst rates or number of blastocysts were observed	Increased sperm oxidative stress leads to a reduction of embryo quality.
27	Yamauchi et al. 2007 [37]	To test if embryo DNA synthesis is related to paternal DNA degradation in zygotes, which is induced by sperm DNA damage.	Mouse	Sperm DNA damage induction through manganese and calcium incubations	- DNA damage: Pulsed Field Gel Electrohporesis (PFGE)	Treatment caused DNA damage detected through PFGE.	Super-ovulated	ICSI	- Fertilization with vas deferens damaged sperm led to DNA degradation in paternal pronucleous. - Intiation of male and female pronucleus DNA replication was delayed. - Blastocyst development was arrested	Impairment of DNA synthesis in the embryo is related to DNA damage in sperm.
28	Yamauchi et al. 2007 [64]	To test embryo development from sperm cells to which DNA damage was induced.	Mouse	DNA damage induction through manganese and calcium incubations	- DNA damage: Pulsed Field Gel Electrohporesis	Treatment caused DNA damage detected through PFGE.	Super-ovulated	ICSI	- DNA damage caused a reduction in the percentage of embryos reaching two-cell stage - Development to blastocyst was impaired when sperm DNA was damaged	Embryo development is impaired when the fertilizing sperm presents chromatin fragmentation.
29	Yi et al. 2016 [65]	To examine the effects of Davallialactone (DAVA) on in vitro sperm fertilizing ability.	Pig	Spermatozoa were incubated with DAVA	- ROS production: intracellular H_2_O_2_	DAVA reduced intracellular H_2_O_2_ in sperm cells.	IVM	IVF	- Fertilization was enhanced by DAVA	Addition of DAVA to fertilization medium reduces ROS and increases fertilization rates
30	Yi et al. 2017 [66]	To determine the impact of difructose dianhydride IV (DFA-IV) on in vitro sperm fertilizing ability.	Pig	Spermatozoa were incubated in DFA-IV at different concentrations for two hours	- Total oxidative stress.	Total intracellular ROS levels were decreased in samples incubated with DFA-IV.	IVM	IVF	- Higher fertilization rates - Higher cleavage and blastocyst rates	Addition of DFA-IV to sperm increases fertilization and blastocyst rates.

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
