# Peer review of "The Relationship between Sperm Oxidative Stress Alterations and IVF/ICSI Outcomes: A Systematic Review from Nonhuman Mammals"

_biology, 2020, doi:10.3390/biology9070178_

Round 1
Reviewer 1 Report
This is an interesting and useful review on the effect of the oxidative stress on spermatozoa and in vitro techniques outcomes. The authors carefully revise the literature, highlighting relevant information. Whereas the manuscript is well organised and written, it could benefit from some text editing and some justification (at least by the abstract/introduction) on why choosing to perform this review on non-human species, if focused on human ART.
The text requires some careful editing though. For instance, in the abstract, "After IVF/ICSI treatments...studies observed a *positive* relationship between sperm oxidative stress and fertilisation rates...", but indeed, "In conclusion, the present systematic review shows that sperm oxidative stress is associated with a significant reduction in fertilisation rates and in vitro embryo development." Either this is a mistake or maybe the wording should be changed.
In the abstract, maybe the authors should highlight the usefulness of carrying the review on non-human mammals (they skip directly from stating the problem to explaining the content, with no additional justification).
Whereas the text is easy to follow, the readability could benefit from some revision for removing unnecessary words/expressions, avoiding the passive voice, splitting long sentences, etc. Some parts are apparently contradictory or at least could gain clarity. For instance, in 55-57, there is a false opposition, it could be rewritten as "These figures show that there is still room for the improvement of these methods. Therefore, it is necessary to gain new insights into the factors that allow the production of embryos with high quality and implantation chances."
As in the abstract, the introduction lacks a firm justification of why writing a background on human ART and then deciding to perform a review on non-human mammals. The reader miss some reasoning by line 99 connecting this two parts.
In results, the authors make an excellent work summarising the findings, with a very complete display in Table 1. Whereas the description in 3.2 is clear, I wonder if some of that could be summarised in one or two contingency tables (especies vs. methods: ART, oxidative treatments, etc.).
In 245, "On the other hand..." lacking.
Author Response
Dear Editor and Reviewers,
Enclosed please find the revised manuscript #biology-869947 entitled “The relationship between sperm oxidative stress alterations and IVF/ICSI outcomes: a systematic review from non-human mammals” which we would like to be reconsidered for publication in Biology (MDPI).
A list of detailed answers to the reviewers has been included below. We thank all of them for their helpful and thoughtful critique of our manuscript.
We sincerely thank the Reviewers for the global appreciation of the submitted manuscript, as well as for the opportunity that our manuscript is reviewed in Biology (MDPI) and all the valuable and constructive comments on the first version of the manuscript. We have tried to address in detail and accordingly all the concerns and questions from the reviewers. We provide itemized responses to each point aroused by each of the Reviewers and included the changes (MS track changes) and comments in the manuscript, when required.
----
Reviewer 1
Authors’ response: We sincerely thank Reviewer #1 for the global appreciation of our manuscript, as well as for all the valuable comments and suggestions provided in the following lines, which have greatly improved the first version of this manuscript. We have addressed all of them in each of the following points, as well as in the manuscript, when required. Please find below the itemized responses to all Reviewer #1’s comments.
Reviewer: This is an interesting and useful review on the effect of the oxidative stress on spermatozoa and in vitro techniques outcomes. The authors carefully revise the literature, highlighting relevant information. Whereas the manuscript is well organised and written, it could benefit from some text editing and some justification (at least by the abstract/introduction) on why choosing to perform this review on non-human species, if focused on human ART.
Authors’ response: The controversy regarding the effect of male oxidative stress to IVF/ICSI cycles has been active for years. Regarding this, different narrative and systematic reviews have been conducted in humans, usually with inconclusive results in the topic due to the multiple male and female factors affecting infertile couples. The rationale of performing a review in non-human subjects is that researchers are able to use animals with proven fertility (both males and females) and specifically control the paternal damage caused by oxidative stress in order to make an a priori correction for potential bias. This helps to the interpretation of the translation of this damage to embryo development.
As suggested by the Reviewer, we included this rationale in the Abstract and Introduction.
Reviewer: The text requires some careful editing though. For instance, in the abstract, "After IVF/ICSI treatments...studies observed a *positive* relationship between sperm oxidative stress and fertilisation rates...", but indeed, "In conclusion, the present systematic review shows that sperm oxidative stress is associated with a significant reduction in fertilisation rates and in vitro embryo development." Either this is a mistake or maybe the wording should be changed.
Authors’ response: We thank the Reviewer for identifying this concrete mistake. We corrected it at the Abstract section. Moreover, a proofreading of all the manuscript has also been performed.
Reviewer: In the abstract, maybe the authors should highlight the usefulness of carrying the review on non-human mammals (they skip directly from stating the problem to explaining the content, with no additional justification).
Authors’ response: As stated above, we included this rationale in Abstract and Introduction.
Reviewer: Whereas the text is easy to follow, the readability could benefit from some revision for removing unnecessary words/expressions, avoiding the passive voice, splitting long sentences, etc. Some parts are apparently contradictory or at least could gain clarity. For instance, in 55-57, there is a false opposition, it could be rewritten as "These figures show that there is still room for the improvement of these methods. Therefore, it is necessary to gain new insights into the factors that allow the production of embryos with high quality and implantation chances."
Authors’ response: We completely agree with this comment. This clarifies the concept and improves the readability. As stated above, we proofread all the manuscript to improve readability.
Reviewer: As in the abstract, the introduction lacks a firm justification of why writing a background on human ART and then deciding to perform a review on non-human mammals. The reader miss some reasoning by line 99 connecting this two parts.
Authors’ response: In agreement to the Reviewer, we added a short paragraph connecting these two parts.
Reviewer: In results, the authors make an excellent work summarising the findings, with a very complete display in Table 1. Whereas the description in 3.2 is clear, I wonder if some of that could be summarised in one or two contingency tables (especies vs. methods: ART, oxidative treatments, etc.).
Authors’ response: As suggested by the reviewer, we included a summary table in this section.
Reviewer: In 245, "On the other hand..." lacking.
Authors’ response: Corrected as suggested.
----
We do hope that all the modifications we have made to the manuscript will make it possible for the paper to be accepted and published in Biology (MDPI).
We are looking forward to receiving the editorial decision concerning the new submitted article.
Yours sincerely,
Dr. Jordi Ribas-Maynou and Dr. Albert Salas-Huetos
Dr. Jordi Ribas-Maynou
Biotechnology of Animal and Human Reproduction (TechnoSperm), Institute of Food and Agricultural Technology, University of Girona, 17,003 Girona, Spain. Tel: (+34) 972 419 514, E-mail: j.ribas87@gmail.com
Dr. Albert Salas-Huetos
Andrology and IVF Laboratory, Division of Urology, Department of Surgery, University of Utah School of Medicine, 84180 Salt Lake City, UT, USA. Tel: (+1) 385 210 5534, E-mail: albert.salas@utah.edu

Reviewer 2 Report
This is an overall well-written review article summarizing the relationship between sperm oxidative stress and IVF/ICSI outcomes. I have some suggestions to further improve the manuscript by adding the effect of oxidative stress in sperm epigenome and sperm-mediated epigenetic inheritance, which is is a raising topic and should be discussed. For example, cigarette smoke induced oxidative stress induce sperm DMRs and offspring phenotype, (PLoS Genet 2020, PMID:32520939). Also, oxidative stress is a strong inducer of tRNA cleavage and will generate more tsRNA and rRNA fragments (RNA 2018, PMID:18719243), and both tRNA and rRNA fragments (tsRNAs and rsRNAs) have been involved in sperm mediated intergenerational epigenetic inheritance of phenotypes (Cell Res 2012, PMID 23044802; Science 2016, PMID:26721680; Science 2016, PMID:26721685; Nat Cell Biol 2018, PMID:29695786). There are further evidence showing a close link between sperm mitochondrial function, oxidative stress and sperm tsRNA/rsRNA biogenesis affected by human high-sugar diet (PLoS Biol 2019, PMID:31877125; Nat Rev Endocrinol 2020, PMID:32066893). Sperm tsRNAs/rsRNAs profile can also be used as biomarker for discriminating sperm quality in reard to embryo viability (Cell Discov 2019, PMID: 30992999). Moreover, oxidative stress provide a link between histone modifications and tsRNA biogenesis in testis and sperm that contribute to epigenetic inheritance (Mol Cell 2020, PMID: 32197065; Mol Cell 2020, PMID: 32386538). These are interesting clues that should be discussed in the paper as this point to a future research direction of oxidative stress in sperm biology.
Author Response
Dear Editor and Reviewers,
Enclosed please find the revised manuscript #biology-869947 entitled “The relationship between sperm oxidative stress alterations and IVF/ICSI outcomes: a systematic review from non-human mammals” which we would like to be reconsidered for publication in Biology (MDPI).
A list of detailed answers to the reviewers has been included below. We thank all of them for their helpful and thoughtful critique of our manuscript.
We sincerely thank the Reviewers for the global appreciation of the submitted manuscript, as well as for the opportunity that our manuscript is reviewed in Biology (MDPI) and all the valuable and constructive comments on the first version of the manuscript. We have tried to address in detail and accordingly all the concerns and questions from the reviewers. We provide itemized responses to each point aroused by each of the Reviewers and included the changes (MS track changes) and comments in the manuscript, when required.
----
REVIEWER 2
Authors’ answer: We sincerely thank Reviewer #2 for the global appreciation of our study, as well as for all the valuable comments and suggestions provided in the following lines, which have greatly improved the first version of the manuscript. We have addressed all them in the manuscript.
Reviewer: This is an overall well-written review article summarizing the relationship between sperm oxidative stress and IVF/ICSI outcomes. I have some suggestions to further improve the manuscript by adding the effect of oxidative stress in sperm epigenome and sperm-mediated epigenetic inheritance, which is is a raising topic and should be discussed.
For example, cigarette smoke induced oxidative stress induce sperm DMRs and offspring phenotype, (PLoS Genet 2020, PMID:32520939). Also, oxidative stress is a strong inducer of tRNA cleavage and will generate more tsRNA and rRNA fragments (RNA 2018, PMID:18719243), and both tRNA and rRNA fragments (tsRNAs and rsRNAs) have been involved in sperm mediated intergenerational epigenetic inheritance of phenotypes (Cell Res 2012, PMID 23044802; Science 2016, PMID:26721680; Science 2016, PMID:26721685; Nat Cell Biol 2018, PMID:29695786).
There are further evidence showing a close link between sperm mitochondrial function, oxidative stress and sperm tsRNA/rsRNA biogenesis affected by human high-sugar diet (PLoS Biol 2019, PMID:31877125; Nat Rev Endocrinol 2020, PMID:32066893). Sperm tsRNAs/rsRNAs profile can also be used as biomarker for discriminating sperm quality in reard to embryo viability (Cell Discov 2019, PMID: 30992999).
Moreover, oxidative stress provide a link between histone modifications and tsRNA biogenesis in testis and sperm that contribute to epigenetic inheritance (Mol Cell 2020, PMID: 32197065; Mol Cell 2020, PMID: 32386538).
These are interesting clues that should be discussed in the paper as this point to a future research direction of oxidative stress in sperm biology.
Authors’ answer: We would like to thank the reviewer for the manuscript document. We acknowledge her/his contribution and we have included a paragraph regarding epigenetics and oxidative stress, as suggested. The references suggested have also been included.
----
We do hope that all the modifications we have made to the manuscript will make it possible for the paper to be accepted and published in Biology (MDPI).
We are looking forward to receiving the editorial decision concerning the new submitted article.
Yours sincerely,
Dr. Jordi Ribas-Maynou and Dr. Albert Salas-Huetos
Dr. Jordi Ribas-Maynou
Biotechnology of Animal and Human Reproduction (TechnoSperm), Institute of Food and Agricultural Technology, University of Girona, 17,003 Girona, Spain. Tel: (+34) 972 419 514, E-mail: j.ribas87@gmail.com
Dr. Albert Salas-Huetos
Andrology and IVF Laboratory, Division of Urology, Department of Surgery, University of Utah School of Medicine, 84180 Salt Lake City, UT, USA. Tel: (+1) 385 210 5534, E-mail: albert.salas@utah.edu

Reviewer 3 Report
This review is original in the sense that most review on the subject focused on human subjects and not on non human mammals (mice, cattle, macaques, hamster, pig, rat, sheep). The authors filtered drastically the studies to keep 30 research papers only starting from 3642 papers. Is this strong selection essentially due to the studies based on human patients? There should be some studies where a combination of mice and humans were used; if this is the case, how many of the papers were excluded for this type of reasons?
The authors conclude that a large proportion of studies yield actual effects; however, there may be biases of non-publication of negative studies, a classical reason for finding associations where there is none.
In this type of metaanalysis, the presentation is generally given by a forest plot, which summarizes very well the results of the different studies. Why did not the authors use this type of synthetic representation. Besides, there are softwares for testing the homogeneity/heterogeneity of the data between the different papers (see for instance Kulinskaya et al, Biometrics, 2011)
Author Response
Dear Editor and Reviewers,
Enclosed please find the revised manuscript #biology-869947 entitled “The relationship between sperm oxidative stress alterations and IVF/ICSI outcomes: a systematic review from non-human mammals” which we would like to be reconsidered for publication in Biology (MDPI).
A list of detailed answers to the reviewers has been included below. We thank all of them for their helpful and thoughtful critique of our manuscript.
We sincerely thank the Reviewers for the global appreciation of the submitted manuscript, as well as for the opportunity that our manuscript is reviewed in Biology (MDPI) and all the valuable and constructive comments on the first version of the manuscript. We have tried to address in detail and accordingly all the concerns and questions from the reviewers. We provide itemized responses to each point aroused by each of the Reviewers and included the changes (MS track changes) and comments in the manuscript, when required.
----
REVIEWER 3
Authors’ response: We sincerely thank Reviewer #3 for the global appreciation of our study, as well as for all the valuable comments and suggestions provided in the following lines, which have greatly improved the first version of the manuscript. We have addressed all them in each of the following points, as well as in the manuscript, when required. Please find below the itemized responses to the comments.
Reviewer: This review is original in the sense that most review on the subject focused on human subjects and not on non human mammals (mice, cattle, macaques, hamster, pig, rat, sheep). The authors filtered drastically the studies to keep 30 research papers only starting from 3642 papers. Is this strong selection essentially due to the studies based on human patients?
Authors’ response: Thank you for your comments. This strong selection has been performed in two steps, as stated in section 2.4. First, researchers selected articles based on the title and the abstract; thereafter, the selected studies were assessed by full text. The screening stage led to 80 articles selected for the evaluation of their full text, whereas the second step (eligibility stage) led to 30 articles selected.
The strong selection at the first stage (based on title and abstract) was due to the presence of studies not related to male infertility evaluation, studies based on human subjects, and studies with outcomes different than those the fell into the scope of the study (ART outcomes).
The selection at the second stage, as defined at Figure 1, consisted of an eligibility process with concrete reasons of exclusion. Detailed inclusion/exclusion criteria can be consulted in Supplementary Table 1 (PICOS design structure) as Preferred Reporting Items for Systematic Reviews and Meta-Analyses (PRISMA) guidelines suggested.
Reviewer: There should be some studies where a combination of mice and humans were used; if this is the case, how many of the papers were excluded for this type of reasons?
Authors’ response: The systematic search led to five studies that performed tests combining human sperm with hamster results. Since the aim of the present review was to select those studies performed in animals where sperm damage was induced (seeking homogeneity of the samples), we did not include such type of studies.
Reviewer: The authors conclude that a large proportion of studies yield actual effects; however, there may be biases of non-publication of negative studies, a classical reason for finding associations where there is none.
Authors’ response: We agree to the reviewer regarding publication bias and, thus, we included that point as a limitation in the Discussion (section 4.1).
Reviewer: In this type of metaanalysis, the presentation is generally given by a forest plot, which summarizes very well the results of the different studies. Why did not the authors use this type of synthetic representation. Besides, there are softwares for testing the homogeneity/heterogeneity of the data between the different papers (see for instance Kulinskaya et al, Biometrics, 2011)
Authors’ response: The reviewer is correct that, for meta-analysis, a forest plot is the best way to show the outcome of a specific parameter including different studies.
In our study, we performed a systematic review without the quantitative analysis of data (meta-analysis), so this type of analysis is not suitable in our case. The reason that we couldn’t perform a meta-analysis is because the induction of oxidative damage was not performed under similar conditions across studies, and therefore, studies could not be grouped in a single category of cause – effect (see section 3.2: Effect of the different treatments to spermatozoa). Because of that, we performed an objective, comprehensive and careful review of all the high-quality included studies and we summarized data in an exhaustive table. From this table, we could extract the conclusions of each study regarding IVF/ICSI outcomes (see section 3.2: Effect of the different treatments to IVF/ICSI outcomes)
----
We do hope that all the modifications we have made to the manuscript will make it possible for the paper to be accepted and published in Biology (MDPI).
We are looking forward to receiving the editorial decision concerning the new submitted article.
Yours sincerely,
Dr. Jordi Ribas-Maynou and Dr. Albert Salas-Huetos
Dr. Jordi Ribas-Maynou
Biotechnology of Animal and Human Reproduction (TechnoSperm), Institute of Food and Agricultural Technology, University of Girona, 17,003 Girona, Spain. Tel: (+34) 972 419 514, E-mail: j.ribas87@gmail.com
Dr. Albert Salas-Huetos
Andrology and IVF Laboratory, Division of Urology, Department of Surgery, University of Utah School of Medicine, 84180 Salt Lake City, UT, USA. Tel: (+1) 385 210 5534, E-mail: albert.salas@utah.edu
